# Identification of Genetic Markers and Genes Putatively Involved in Determining Olive Fruit Weight

**DOI:** 10.3390/plants12010155

**Published:** 2022-12-29

**Authors:** Martín Moret, Jorge A. Ramírez-Tejero, Alicia Serrano, Elena Ramírez-Yera, María D. Cueva-López, Angjelina Belaj, Lorenzo León, Raúl de la Rosa, Aureliano Bombarely, Francisco Luque

**Affiliations:** 1Departamento de Biología Experimental, Instituto Universitario de Investigación en Olivar y Aceites de Oliva, Universidad de Jaén, 23071 Jaén, Spain; 2Centro de Investigación y Formación Agraria de Alameda del Obispo, Instituto de Investigación y Formación Agraria y Pesquera (IFAPA), 14004 Córdoba, Spain; 3Instituto de Biología Molecular y Celular de Plantas (IBMCP), CSIC and Universitat Politécnica de Valencia, 46011 Valencia, Spain

**Keywords:** olive, fruit development, fruit size, genetic markers, GWAS

## Abstract

The fruit size of a cultivated olive tree is consistently larger than its corresponding wild relatives because fruit size is one of the main traits associated with olive tree domestication. Additionally, large fruit size is one of the main objectives of modern olive breeding programs. However, as the long juvenile period is one main hindrance in classic breeding approaches, obtaining genetic markers associated with this trait is a highly desirable tool. For this reason, GWAS analysis of both genetic markers and the genes associated with fruit size determination, measured as fruit weight, was herein carried out in 50 genotypes, of which 40 corresponded to cultivated and 10 to wild olive trees. As a result, 113 genetic markers were identified, which showed a very high statistically significant correlation with fruit weight variability, *p* < 10^−10^. These genetic markers corresponded to 39 clusters of genes in linkage disequilibrium. The analysis of a segregating progeny of the cross of “Frantoio” and “Picual” cultivars allowed us to confirm 10 of the 18 analyzed clusters. The annotation of the genes in each cluster and the expression pattern of the samples taken throughout fruit development by RNAseq enabled us to suggest that some studied genes are involved in olive fruit weight determination.

## 1. Introduction

The cultivated olive tree (*Olea europaea* L.) belongs to the Oleaceae family. Although several authors have discrepancies with the specific area where this crop was first domesticated, it is assumed that it originates from the Middle East [1]. As its cultivated area is so large worldwide (10 million hectares), olive grove cultivation has a huge socio-economic impact. The presence of this crop in the entire planet is not uniform because 98% of cultivated hectares with olive groves are found in the Mediterranean Basin, 1.2% in the American continent, 0.4% in East Asia, and the rest in Oceania.

The long recursive contraction and expansion processes of wild populations produced by many climate changes, the limited flow of genes imposed by geographical distance and natural barriers, such as deserts and seas, are some of the factors that have shaped their geographical dissemination and genetic structure [2].

Olive tree domestication is characterized by the vegetative propagation of the most valuable genotypes [3]. They are selected for their agronomic value, such as bigger fruit yields, larger fruit size or higher oil content, their ability to grow in anthropogenic environments and the ease with which they can be vegetative-propagated through cuttings or grafts.

The earliest use of wild olive fruit dates back to the Paleolithic [4] and it was not collected until the early Neolithic [5]. The first olive oil extraction dates back to the Copper Age on the Carmel coast [6], and was utilized mainly as ointments in religious rituals. Its culinary use was not recorded until Roman times and reaches our days [7].

Fruit size is one of the main differentiating characteristics between wild and cultivated olive trees [8,9]. The fruit of cultivated olive trees is consistently larger than that of its corresponding wild relatives. The fruit of olive trees is small, from 1 to 4 cm long with a diameter between 0.6 and 2 cm. Thus, fruit fresh weight below 1 g is usually found on wild olives compared to the higher values of cultivated ones [8]. Wild fruit size is restricted by the main mechanism of seed dissemination; that is, the oral cavity of frugivorous birds that ingest it whole. This means that wild fruit cannot exceed a certain size [10].

The cultivated olive tree (*Olea europaea* subsp. *europaea* var. *europaea*) derives from the wild olive tree (*Olea europaea* subsp. *europaea* var. *sylvestris*) and emerged after grain agriculture during the Neolithic [11,12]. Olive tree domestication began with the selection, vegetative propagation, and growing of outstanding wild genotypes [1]. Population studies performed by means of SSR sequences suggest that current olive cultivars are the result of selecting plants produced spontaneously by the cross between cultivars introduced by colonizers and local olive populations [13]. Fruit size and oil content were the main traits associated with olive tree domestication, and they are still of much interest in today’s olive breeding programs [14]. Larger fruit size is extremely important in cultivated olive trees to facilitate harvesting [15]. Therefore, fruit size in traditional olive cultivars is much bigger than in wild ones [16]. However, as the long juvenile period is a main hindrance in classic breeding approaches [17], obtaining genetic markers (GM) to be used in breeding programs for this trait is a highly desirable tool.

In this work, we attempted to find GMs that can facilitate the genetic improvement of fruit size and, more specifically, fruit weight. Genome-wide association studies (GWAS) are becoming a powerful tool for detecting the quantitative trait locus (QTL) associated with agronomic traits of different plant species [18,19,20,21]. To date however, very few GWAS studies are available about olive [22]. To carry out the GWAS analysis we used the “Picual” genome [23], and to determine the chromosome location the wild genome [24]. The present study applies a combination of strategies, including the GWAS analysis, segregating progeny analysis, and RNAseq to define GMs of olive fruit size to be employed by breeders for fruit improvement purposes.

## 2. Results

### 2.1. GMs Obtained by GWAS of Fruit Weight

Only the highly significant variant positions were selected as putative GMs (−log_10_*p* value above 10) for the GWAS analysis. However, a large number of 113 putative GMs were found to be associated with this trait (Figure 1). Many of these putative GMs were clustered in a very short distance. In those cases, in which segregation was observed, the clustered GMs behaved as haplotypes that were inherited as a block (Table 1). Therefore essentially, every cluster can be represented by a single GM to be used in the selection of this trait. Thirty-eight clusters were defined, which suggests that fruit size is a polygenic trait.

### 2.2. Analysis of GMs’ Segregation in a Phenotyped Progeny

The segregation of some of the GMs defined by the GWAS analysis was studied in a “Frantoio” x “Picual” progeny. Female parent “Frantoio” produces smaller fruit than the male “Picual” (Appendix A). Forty selected descendants with extreme phenotypic values (Table 2) were genotyped to determine allele segregation, which could be independent of or linked with the phenotype. Some of the GMs obtained by GWAS did not segregate in this progeny and therefore, could not be further analyzed in the present study. However, the 31 GMs that clustered in 16 different groups were obtained in the progeny (Table 1). Thus, by using a *p*-value of <0.10, 18 out of the 31 GMs were found to segregate with fruit size (Appendix A). The GMs linked with fruit size/weight clustered in nine groups (Table 1; Figure 2), which probably represent nine genes involved in fruit size determination. The remaining 13 GMs did not segregate with this trait and were clustered in seven groups (Table 1; Figure 3). Finally, our results showed that more than half the putative analyzed GMs found in the GWAS analysis were confirmed as real GMs of olive fruit size.

### 2.3. Inheritance Model

The finding of GM clusters and the confirmation of more than half of them by the segregating progeny indicate that this is a polygenic trait. Another important fact was determining if there was an additive inheritance model or if it was due mainly to dominant alleles. This can be determined only if the three possible genotypes, the two types of homozygous, and the heterozygous plants are present in the progeny. Four GMs produced the three possible genotypes in the segregating progeny. Therefore, the inheritance model could be tested in those GMs. In Figure 2, GM2874 [A > G] and GM3346 [G > A] showed a clear and complete dominance of one of the alleles. GM4878 [T > C] seemed to present complete dominance, but one of the homozygous genotypes contained a very small number of trees. This means that it could also fit as incomplete dominance. Finally, GM3361 [A/G] looked like a case of incomplete dominance, but it was not conclusive and could also be compatible with a complete dominance inheritance model. For most GMs, the inheritance model could not be established, but complete dominance seemed frequent when determining this trait. Most GMs were represented by two genotypes, one was homozygous and the other heterozygous. Therefore, phenotypic differences were not detectable when the homozygous genotype had the dominant allele. This means that unconfirmed GM clusters should not be ruled out as good GMs and further studies are required with other progenies to clarify this fact.

### 2.4. Putative Genes Associated with the Fruit Size Trait

For each cluster of GMs confirmed by the segregation progeny, there should be a gene in LD responsible for fruit size differences between alleles. While searching for these genes, we considered the distance of the gene to the GM, the gene annotation, and the expression profile during fruit development, and from flowers to mature fruit (Figure 4), which was studied by RNA-seq. Regarding the distance to the GM, only the genes found in LD by using the TASSEL software were considered. In some clusters, the distances of the genes apparently in LD were too long to be convincing because distances of hundreds of kbases are hardly believable to be in LD. According to the TASSEL analysis, these long distances in LD were probably the result of a limited number of genotypes and lack of information about their family relationship, if it indeed existed.

According to our data, GMs at a distance of around 25 kbases were inherited as haplotypes in the GM0091 cluster. Therefore, the probability of those genes at longer distances than 25 kbases to the GM cluster being responsible for trait variability was considered low. In clusters GM0029B and GM0029C, distances of seven kbases seemed adequate for producing recombinant genotypes in the progeny (Appendix A). Regarding the expression profile, all the genes not expressed at any time during the fruit development process were ruled out for being responsible for fruit size determination. The nine confirmed GM clusters linked with fruit size/weight are discussed below.

#### 2.4.1. GM0091

This cluster included 15 GMs that cover 32 kbases and were inherited as a haplotype. The TASSEL analysis produced 15 genes in LD with the GMs of this cluster. Only five of those genes were expressed during fruit development. Of them, Oleur061Scf0091g03021.1 is a gene that codes a calmodulin-binding protein (DUF1645) that has been related to drought tolerance [25]. We have found that its expression increases in summer. Oleur061Scf0091g04008.1 is a gene that codes for a 2-oxoglutarate (2OG) and Fe(II)-dependent oxygenase superfamily protein involved in ethylene formation and anthocyaninidin biosynthesis that is induced during fruit maturation. Oleur061Scf0091g04023.1 is an Early Flowering MYB (AT2G03500.1) transcription factor that acts as a flowering repressor. Oleur061Scf0091g04027.1 is a gene with homology to AT2G26520.1 that codes a weakly expressed transmembrane protein in the first month of development. Finally, Oleur061Scf0091g04020.1 codes an Armadillo repeat-containing protein 6 involved in regulating plant development and signaling [26]. This gene was induced for the first 15 days of fruit development and remained high until the last month, when its expression lowered to a similar level as that in flowers (Figure 5). Accordingly, this gene seemed to be the most probable one to determine fruit size in LD with this GM0091 cluster.

#### 2.4.2. GM0306

This cluster included four GMs and, according to TASSEL, there were 23 genes in LD with them. Only eight of these genes were expressed during fruit development and four of them were hundreds of kbases away from GMs. So, they were unlikely to be found in a real LD with the GM0306 cluster. Gene Oleur061Scf0306g06001.1 was placed over GM0306 and coded for DNA polymerase III, subunit gamma/tau, P-loop containing nucleoside triphosphate hydrolase in bacteria, and was also present in plants, but with no clear function [27,28]. It was expressed in flowers and very early fruit development steps to be repressed after the second month of fruit development (Figure 5). Therefore, it could promote DNA replication in the early developing fruit stage, a time when cell division activity is considerable. Another gene was S-adenosyl-L-methionine-dependent methyltransferase (Oleur061Scf0306g06010.1), which is homologous to AT1G24480.1, could be involved in cell differentiation and development growth. It was highly expressed in flowers, which lowered to be repressed from the second month and be re-induced at 6 months of development. Gene differentiation was more likely to be important immediately after full flowering and ovule fecundation in the first rapid cell division stages of fruit, although this was not the case of this gene. Therefore, it was less likely to be a determinant for fruit size. The other two genes, Oleur061Scf0306g06016.1 and Oleur061Scf0306g06017.1, coded for an initiation factor eIF-4 gamma and a tetratricopeptide-like helical domain, but did not seem to be good candidates to control fruit size.

#### 2.4.3. GM2874

According to the TASSEL analysis, GM2874 was in LD with 84 genes that expanded over 1.5 Mbases. As this made no biological sense, we focused on those genes no further than 80 kbases from GM2874. Only one gene was expressed during fruit development (Oleur061Scf2874g08017.1). This gene was around 20 kbases away from GM2874, which came close enough to be in a real LD and was highly expressed in very early fruit development steps. Expression lowered in later stages, especially at 5 and 6 months (Figure 5). According to the Arabidopsis Information Resource (TAIR), this gene coded for an AT5G13100—gap junction beta-4 protein, which is involved in many processes, including cell division, developmental growth, the hormone-mediated signaling pathway, and plant-type cell wall biogenesis. Its high expression in early developing fruit was consistent with the marked cell division activity in this phase.

#### 2.4.4. GM3346

In this case, the TASSEL analysis did not produce any genes in the LD with GM3346, except Oleur061Scf3346g05040.1, which contained the GM. This gene coded for glycosyl transferase, family 17. According to the TAIR (AT1G12990), this protein is involved in a number of processes; for instance, anatomical structure maturation, developmental growth, the hormone-mediated signaling pathway, plant epidermis development, root morphogenesis, among others. This gene was expressed all the time during fruit development. It was highly expressed in flowers, followed by lower expression levels during most fruit development and was highly expressed again at 5 months of fruit growth (Figure 5).

#### 2.4.5. GM3361

The TASSEL analysis found 37 genes in LD with this GM cluster, covering nearly 500 kbases. When we analyzed only the genes that were no further than 80 kbases, four genes were expressed within that range during fruit development. Three of them performed functions that did not seem to be likely involved in fruit size determination. Thus, according to the TAIR, Oleur061Scf3361g05012.1 coded for a homolog of a heavy metal transport/detoxification superfamily protein, Oleur061Scf3361g05013.1 coded for an ubiquitin ligase that regulated amino acid export, and Oleur061Scf3361g06003.1 coded for REF4-related 1, which was involved in phenylpropanoid metabolic process regulation. Finally, Oleur061Scf3361g05004.1 coded for a cytochrome P450 family protein involved in cell differentiation, developmental growth, the response to a nitrogen compound, root development and tissue development. This last gene was expressed upon full flowering, its expression increased from the beginning of fruit development and then dropped to low levels at the end of fruit development at 6 months (Figure 5). For this reason, this gene seems to be a good candidate to be involved in determining fruit size.

#### 2.4.6. GM3663

The result of the LD with TASSEL produced 24 genes at distances longer than 200 Mbases from GMs. When focusing on the genes no further than 80 kbases, only six genes were found to be expressed during the fruit development process. Three of these genes performed functions that did not seem relevant for determining fruit size. Oleur061Scf3663g04007.1 coded for an aminopeptidase, Oleur061Scf3663g04009.1 coded for a class II heat shock protein, and Oleur061Scf3663g04031.1 coded for an unknown function protein with a low expression in both flowers and fruit. However, the other three genes could be candidates to participate in fruit development and for determining fruit size. Thus Oleur061Scf3663g03006.1 coded for a serine/threonine kinase, was expressed in flowers, was significantly induced at 15 days since full flowering and its high expression remained until the last 2 months of fruit growth (Figure 5). Oleur061Scf3663g04026.1 coded for a Pleckstrin homology domain superfamily protein with homology to AT2G30060 involved in cell wall biogenesis, developmental growth, plant-type cell wall organization or biogenesis, the protein catabolic process, the response to cadmium ion, and the response to inorganic substance. This gene was expressed in flowers, its expression increased early from 15 days after flowering and it remained high until the end of fruit development (Figure 5). This could probably be another good candidate to be involved in fruit size determination. Finally, Oleur061Scf3663g04027.1 coded for an mRNA splicing factor, thioredoxin-like U5 snRNP. It was expressed in flowers and its expression lowered from the first month to the end of the fruit development process (Figure 5). Therefore, this gene does not seem to be a good candidate to determine fruit size, but it cannot be ruled out.

#### 2.4.7. GM4878

In this cluster, the LD analysis produced six genes at distances shorter than 60 kbases. Only one gene was expressed during fruit development. This Oleur061Scf4878g00020.1 gene coded for an lncRNA, which was also present in the wild olive genome [24]. Similar sequences were found, but only in the closer genus of *Fraxinus* and not in any other organism. This lncRNA was slightly expressed in flowers and its expression began to increase from the very early fruit development steps with a high expression level after 1 month from flowering, which remained high until the development process ended (Figure 5).

#### 2.4.8. GM5641

Seven genes were found in LD with this GM and covered distances shorter than 70 kbases. Five of these genes (Oleur061Scf5641g00017.1, Oleur061Scf5641g01009.1, Oleur061Scf5641g01037.1, Oleur061Scf5641g01039.1, Oleur061Scf5641g01045.1) were expressed during fruit development. All of them started from different expression levels in flowers, were induced 15 days after flowering and their high expression remained until fruit development ended (data not shown). Their different functions did not clarify which of them could be involved in fruit size determination.

#### 2.4.9. GM6972

Six genes were found in LD with this GM by TASSEL, with all of them at shorter distances than 60 kbases. Only two genes were expressed during fruit development. Oleur061Scf6972g00012.1 coded for an RAB6-interacting golgin protein with an unclear function, but related to the stress response in wheat and Arabidopsis [29]. Therefore, it could not be taken as a good candidate gene for determining fruit size. On the contrary, gene Oleur061Scf6972g00009.1 could be considered a good candidate because it coded for fructose-bisphosphate aldolase, class-I, involved in several processes, such as the fructose 1,6-bisphosphate metabolic process, gluconeogenesis, the glycolytic process, and the mitochondria-nucleus signaling pathway. This gene was highly expressed in both flowers and fruit, and its expression profile was significantly high between months 2 and 5. Its expression peaked by months 3 and 4 (Figure 5). The period when this gene’s expression profile was very high coincided with that of increased cell size in developing fruit [30].

## 3. Discussion

GWAS analysis is expected to facilitate olive genetic breeders’ work. In this context, a first GWAS analysis of four olive traits has been recently described [22]. In this work, we ran a GWAS analysis to look for GMs to be used to select olive genotypes with a larger/heavier fruit size. The GWAS analysis was run with the genomic data of 50 genotypes, 40 cultivated olive varieties and 10 wild olive ones [23], and also with the phenotypic data from the WOGBC database [31,32] and the wild phenotypes obtained in this work (Appendix A). Although 50 accessions may seem scarce for a GWAS analysis, they were selected to represent more than 90% of the genetic variability of the cultivated olive tree and a wide geographical representation of the wild olive trees. Furthermore, we analyzed the whole sequenced genomes, a better method indeed than GBS technology for GWAS analysis [33]. This is probably why we obtained GWAS *p*-values 1,000,000-fold smaller than the GM *p*-values in a previous GWAS analysis in olive [22], with a Manhattan plot cut-off about 4 in [22] in contrast to 10 in our work. Even more relevant is that we got experimental confirmation of a number of GMs obtained in the GWAS analysis.

The GWAS analysis produced high-quality genetic variants that could be good GMs for fruit weight. These genetic markers corresponded to 39 clusters of genes in LD. A higher score from 5 to 7 for the −log_10_ pvalue is usually required to be selected as a variant of a possible GM. A recent GWAS about olive set thresholds below 5 [22]. In our work, the threshold was set at 10. With this threshold, a relatively large number of GMs (113) showed very high statistical significance with fruit size variability. Albeit in a lower proportion, significant associations for fruit weight, stone weight and fruit flesh with the pit ratio have been found in previous studies into olive [22]. The number of variants herein selected would have been thousands if we had set the threshold at 9. These results, which are in accordance with previous evaluations of olive progenies [34,35], suggest an unsurprising result about the fruit size phenotype being a polygenic trait.

The 31 GMs obtained in the 16 GM clusters analyzed in the “Frantoio” x “Picual” progeny were found to include genetic variants that are inherited as haplotypes. Only in four GMs could the inheritance model for the recessive/dominance or additive mode be analyzed. At least two of the four GMs presented a recessive/dominance model, and the other two were not completely clear, but a third GM could have a recessive/dominance model and another could follow the additive model. Thus, complete dominance relations seemed frequent in this polygenic trait. The segregating progeny study confirmed the linkage of 9 GM clusters of the 16 that were analyzed (Figure 2). For all the unconfirmed clusters, only two genotypes were observed: one heterozygous and the other homozygous. If these genes presented a recessive/dominance inheritance model, and if the homozygous plant had the dominant allele, no differences between the two genotypes could be expected. For this reason, the seven unconfirmed GM clusters could still be good GMs for determining fruit size. In fact, the number of unconfirmed clusters could be the expected one if the recessive/dominance model was the commonest in the genes that coded for the fruit size trait. All these data indicated that the GWAS analysis produced high-quality GMs, which might be confirmed by future works analyzing more segregating progenies from other genetic crosses.

In order to identify the genes most likely involved in fruit size determination, the genes in LD with the nine GM clusters were determined and their expression profiles were studied by RNA-seq during fruit development. The genes not expressed at any time during fruit development were ruled out as possible candidates involved in the fruit size/weight phenotype determination. The combination of the gene annotation, possible function, and/or role in the process and expression profile was used to establish the gene that was most likely responsible for being involved in determining fruit size in seven of the nine GM clusters. Fruit size depends on cell number and cell expansion. Cell division is relevant for the first 6 weeks, while cell expansion is almost solely responsible for the fruit growth that takes place later [36].

GM0306 and GM2874 seemed to be linked with the genes involved in the first fruit growth stages, as determined by the expression profile and the genes implicated in promoting cell division. The genes that most probably are responsible for this phenotype are: a gene coding for a DNA polymerase III, subunit gamma/tau found in bacteria, and also present in plants [27,28], and another gene that codes for a gap junction beta-4 protein involved in many processes, including cell division, developmental growth, the hormone-mediated signaling pathway and plant-type cell wall biogenesis [37]. This finding is consistent with the marked cell division activity observed for the first weeks of olive fruit development [36].

Four GM clusters as the best candidate genes seemed to play a role in controlling the development process. This was the case of GM0091. This candidate gene coded for an Armadillo repeat-containing protein 6 involved in the regulation of plant development and signaling [23]. GM3346 only appeared in LD with a single gene that coded for a glycosyl transferase, family 17, which is a protein involved in several processes, for instance, anatomical structure maturation, developmental growth, the hormone-mediated signaling pathway, plant epidermis development, and root morphogenesis. GM3361 has a candidate gene that codes for a cytochrome P450 family protein that is involved in several processes, such as cell differentiation, developmental growth, the response to nitrogen compound, root development, and tissue development. Finally, GM4878 is only in LD with a gene expressed during fruit development, and this gene did not code for a protein. It coded for an lncRNA that was quite poorly expressed in flowers, and was induced and highly expressed throughout the fruit development process. It is tentative to propose a regulatory role for this lncRNA in fruit development.

In later fruit development stages, the genes involved in cell expansion were expected [33]. The GM6972 candidate gene coded for fructose-bisphosphate aldolase, which is involved in gluconeogenesis, a process that may be relevant during cell expansion.

Therefore, in this work, good-quality GMs were obtained and could be used for the genetic improving of fruit size/weight in olive. A number of these GMs was validated in a segregating progeny. In addition, the present study also allowed the identification of genes in LD with these GMs, the study of their expression profile, and propose a role in olive fruit development.

Hence, two genes seem to play a role in the initial stage of cell divisions, while four genes might play a role in controlling fruit development, of which one codes for an lncRNA. Finally, another gene can be involved in cell expansion in fruit. These GMs could be the most interesting for developing a marker-assisted selection strategy for breeding large-sized olive fruit.

## 4. Materials and Methods

### 4.1. Plant Materials and Fruit Weight Determination

The present work includes 50 olive genotypes (40 cultivated varieties and 10 wild accessions) that represent a wide range of fruit weight phenotypes and reflect all the geographic olive distribution areas in the Mediterranean Basin [32,38]. The same genotypes have been previously included in genomic studies [23]. The fresh fruit weight data from the cultivated varieties were obtained from former studies from the World Olive Germplasm Bank of Córdoba (WOGBC), Spain [31,32], while those of the wild phenotypes were acquired in the present study (Appendix A).

Fruit characteristics were measured in those trees with sufficient fruit load and in homogenous samples (1 kg) during two harvesting seasons [31]. Three subsamples of around 25 g were randomly selected to measure fruit fresh weight and were dried in a forced-air oven at 105 °C for 42 h to ensure dehydration. Then the dried samples were weighted to determine the fruit dry weight.

Additionally, the dry fruit weight for 200 genotypes descending from a “Frantoio” x “Picual” cross was determined, and 40 genotypes equally representing the 20 highest and the 20 lowest average dry fruit weight values were selected for further analyses (Table 1).

The plant material is stored at the Instituto Universitario de Investigación en Olivar y Aceites de Oliva (Universidad de Jaén, Spain).

### 4.2. GWAS Analysis

For the GWAS study, the available olive genome data corresponding to the 40 olive varieties and the 10 wild genotypes [23] were used.

To first focus on the coding sequences and nearby regions, a reduction in the size of the “Picual” reference genome was carried out. Hence the sequences of the olive genes and their flanking regions (1 kb at both ends) were taken as references. This process was carried out by combining the samtools v.1.3.1 and FastaExtract tools. Once the reduced reference genome was composed, the genetic variants that affected these regions were extracted with VCFtools v.0.1.15. The GWAS study was carried out with this reduced reference genome.

Based on the genotypic and phenotypic data of the cultivated and wild plant materials under study, a GWAS analysis was performed with the PLINK v1.90p software. The genotypic data were collected in the individual genetic variants call file (vcf). The phenotypic data included the mean fresh fruit weight for each studied cultivated and wild accession. The statistical significance to determine the genotype-phenotype association was set at *p* < 10^−10^, with a determination coefficient threshold of 0.5.

### 4.3. Genotyping the Segregating Progeny

DNA was extracted and purified from the leaves of the 40 selected genotypes from the progeny using the Illustra Nucleon PhytoPure kit (GE Healthcare, Chicago, IL, USA). Then the DNA fragments of the 18 clusters containing 31 SNPs/INDELs, which were previously associated with fruit size by GWAS, were amplified by PCR in each tree sample. Amplifications were performed in a final volume of 20 µL consisting of 8 μL of DNA at a concentration of 4.5 ng/μL. Next 10 μL of the iTaqTM Universal SYBR^®^ Green Supermix and 2 μL of primer oligonucleotide mix were introduced and amplified in a CFX96 Real-Time thermocycler (BioRad Hercules, CA, USA) by the Central Research Support Services (SCAI) at the Universidad de Jaén. Primers were designed with the Oligo 7 software (Molecular Biology Insights, Inc. (DBA Oligo, Inc.) COS, USA). The amplification program was set at 95 °C for 5 min as the initial denaturation, followed by 50 cycles of 5 s at 95 °C, 15 s at 55 °C, 45 s at 60 °C, and 3 min at 72 °C as the final extension. Finally, amplification was verified by electrophoresis in 4% agarose gel.

Genotyping was performed by next-generation sequencing (NGS) using the Illumina Novaseq 6000 platform at Novogene (Novogene Co., Ltd. Cambridge, UK). The amplicons of each genotype were pooled for sequencing. The sequences of all the genotypes were obtained by NGS for all 40 trees to be aligned with the reference genome using the command line program bowtie2, which is specific for small-/medium-sized sequences [39]. The alignment files obtained in the sam format were compressed, ordered, and indexed using SAMtools v. 1.3.1 [40]. The genotyping data of each tree were obtained by the combination of the 18 genes, whose genotypic information were taken by combining the extraction of the genotype for each specific position using the VCFtools program [41], and manually reviewing the most doubtful positions by visualizing them in Integrative Genomics Viewer (IGV) [42].

### 4.4. Statistical Analysis

The non-parametric Kolmogorov–Smirnov test was performed for each SNP/INDEL marker in the progeny to determine whether the sample came from a normally distributed population. After checking the normality of samples, the average weight was compared according to the inherited genotype to, thus, study whether the present allele was associated with a statistically differential fruit size in relation to that inherited by those plants containing the other allele. To do this, means were compared by a parametric Student’s *t*-test, assuming the absence of differences between them as the null hypothesis and setting the limit of statistical significance to reject the null hypothesis at α = 0.1. The linkage disequilibrium (LD) was determined by means of TASSEL, comparing each candidate genetic marker to the other SNP/INDELs of the scaffold and taking both r^2^ and D’ into account to determine which of them were in the LD.

### 4.5. Transcriptomic Analysis

The transcriptomic analysis of fruit development was carried out by RNAseq. For the transcriptomics study, flower and fruit samples were obtained from three “Picual” trees growing in the experimental field of the Universidad de Jaén. To reduce environmental variability, samples were collected from closely located trees and south-facing branches at different time points during fruit development, i.e., from the first day of flowering, 15 days later, and every month up to 6 months when olive fruit ripened. Total RNA was extracted with the Spectrum™ Plant Total RNA kit (Merck KGaA, Darmstadt, Germany). PoliA^+^ RNA was purified and sequenced with the Illumina Novaseq 6000 platform at Novogene. At least 50 Gb of the Q30 sequences data were obtained from each biological replicate sample. The RNAseq analysis was performed with DNAstar (ArrayStar 17, Rockville, MD, USA) for the RNA-seq analyses (www.dnastar.com) (accessed on 11 November 2022).

## Figures and Tables

**Figure 1 plants-12-00155-f001:**
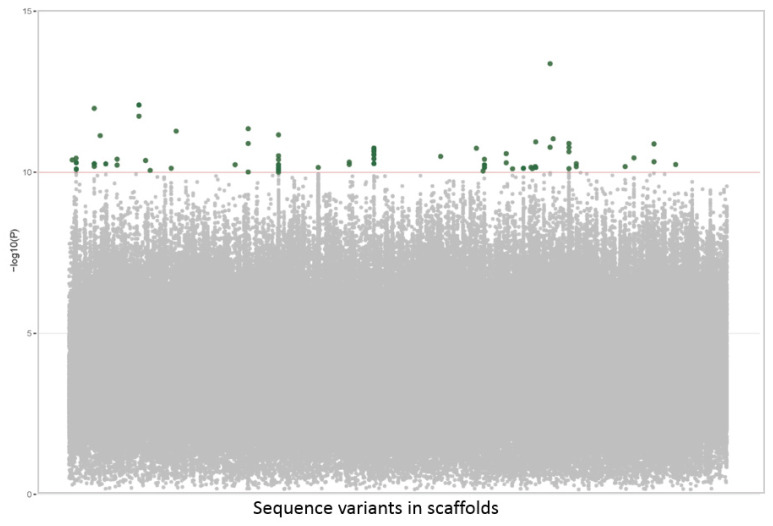
Manhattan plot of the fruit size GWAS results. Variants selected as GM associated with the fruit size phenotype (−log_10_*p* value above 10), green dots. Unselected variants, gray dots.

**Figure 2 plants-12-00155-f002:**
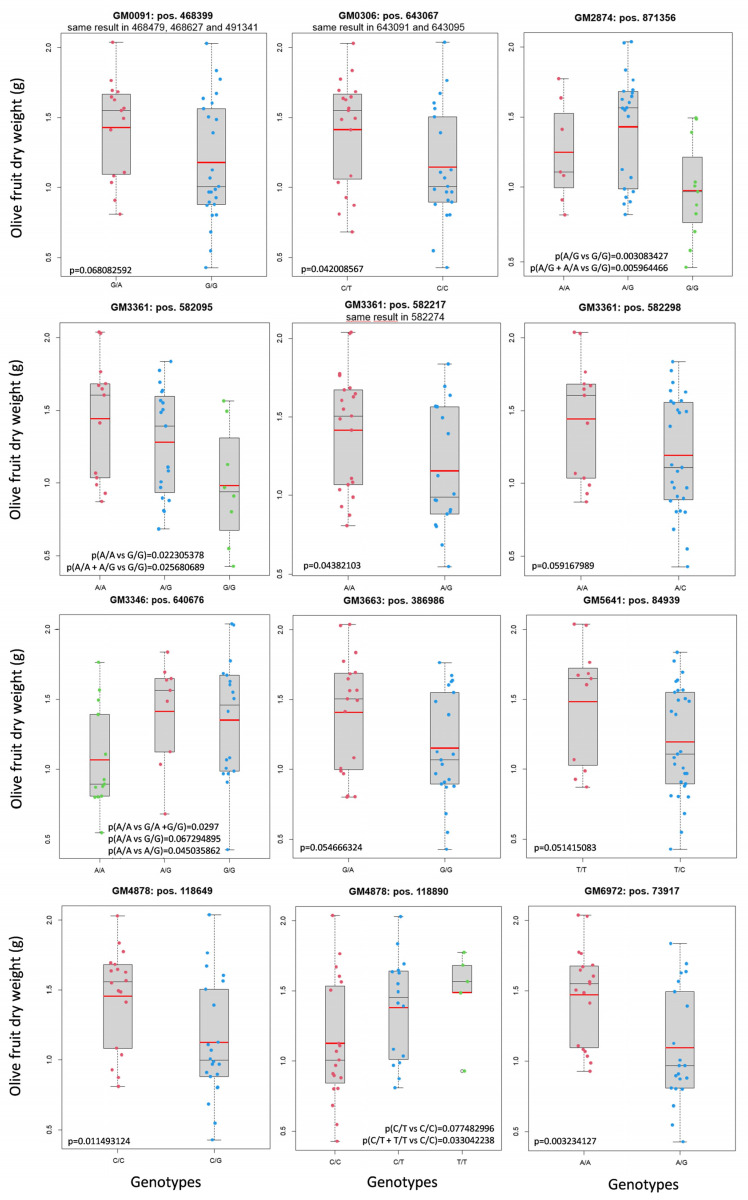
The GM that segregated with the fruit size phenotype. Mean fruit weight, a red line; median, a thin black line.

**Figure 3 plants-12-00155-f003:**
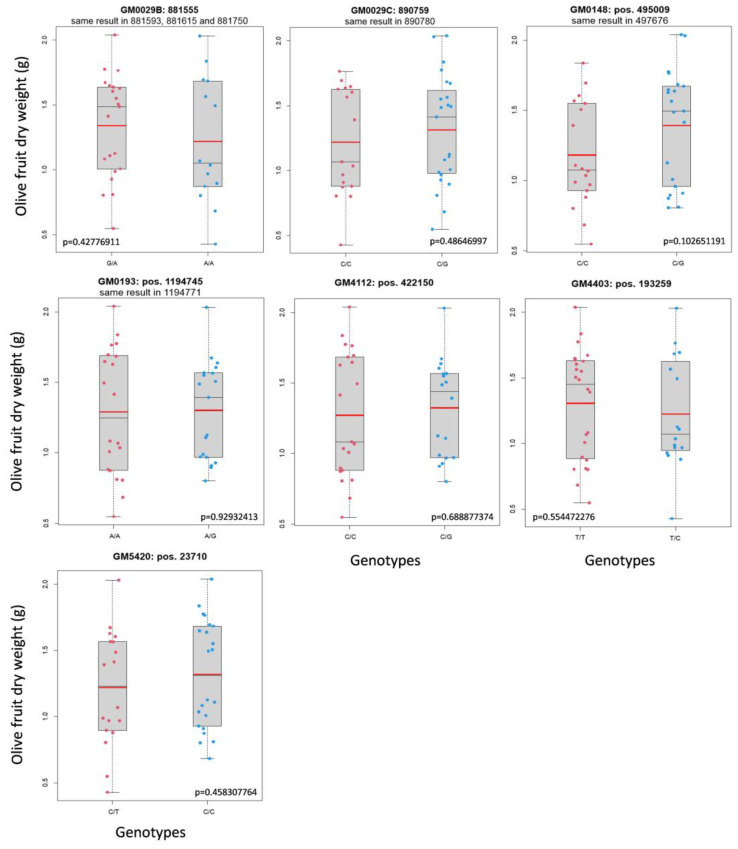
The GM that did not segregate with fruit size. Mean fruit weight, a red line; median, a thin black line.

**Figure 4 plants-12-00155-f004:**
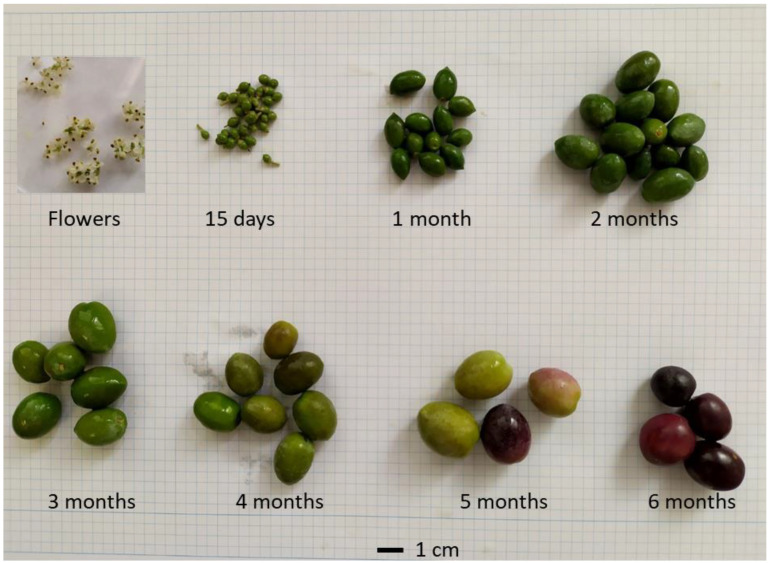
Images of flowers and developing fruits at different sampling times.

**Figure 5 plants-12-00155-f005:**
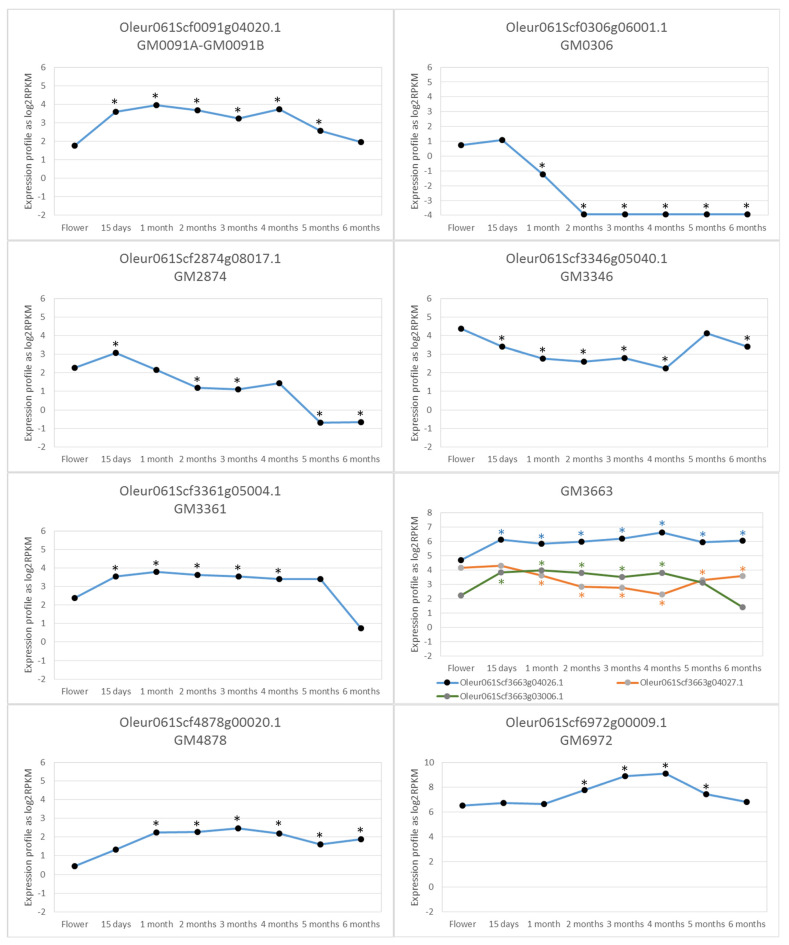
RNAseq gene expression profile of the most probable genes involved in determining the fruit size phenotype at full flowering, 15 days and 1–6 months later. * *p*-value < 0.05.

**Table 1 plants-12-00155-t001:** GMs obtained by GWAS and their segregation in a “Frantoio” x “Picual” progeny.

GM Cluster	Scaffold	GM Positionin Scaffold	Alleles	GM Segregation with the Fruit Size Phenotype	Chromosome *
GM0014	Oleur061Scf0014	300237	A/G	-	18
GM0029A	Oleur061Scf0029	478377	G/T	-	15
GM0029B	Oleur061Scf0029	881555	G/A	NO	15
	Oleur061Scf0029	881593	C/A	NO	15
	Oleur061Scf0029	881615	G/A	NO	15
	Oleur061Scf0029	881750	A/C	NO	15
	Oleur061Scf0029	882112	ATT/AT	-	15
	Oleur061Scf0029	882129	C/T	-	15
GM0029C	Oleur061Scf0029	889252	A/C	-	3
	Oleur061Scf0029	890759	C/G	NO	3
	Oleur061Scf0029	890780	A/G	NO	3
GM0091	Oleur061Scf0091	460570	C/T	-	3
	Oleur061Scf0091	465778	A/G	-	3
	Oleur061Scf0091	465893	A/G	-	3
	Oleur061Scf0091	466037	T/C	-	3
	Oleur061Scf0091	466296	T/C	-	3
	Oleur061Scf0091	468399	G/A	YES	3
	Oleur061Scf0091	468479	C/T	YES	3
	Oleur061Scf0091	468627	A/G	YES	3
	Oleur061Scf0091	468749	T/C	-	3
	Oleur061Scf0091	469150	A/G	-	3
	Oleur061Scf0091	471744	G/T	-	3
	Oleur061Scf0091	491025	G/A	-	1
	Oleur061Scf0091	491341	G/A	YES	1
	Oleur061Scf0091	491618	T/C	-	1
	Oleur061Scf0091	492350	C/G	-	1
GM0122	Oleur061Scf0122	385110	C/T	-	8
GM0148	Oleur061Scf0148	495009	C/T	NO	17
	Oleur061Scf0148	497676	C/G	NO	17
GM0193	Oleur061Scf0193	1194745	A/G	NO	US
	Oleur061Scf0193	1194771	ATTTTTTTG/ATTTTTTA	NO	US
GM0306	Oleur061Scf0306	640199	C/T	-	18
	Oleur061Scf0306	643067	C/T	YES	18
	Oleur061Scf0306	643091	C/A	YES	18
	Oleur061Scf0306	643095	TT/TCT	YES	18
GM0340	Oleur061Scf0340	201854	ATT/GTC	-	6
GM0360	Oleur061Scf0360	830983	A/T	-	US
GM0476	Oleur061Scf0476	81329	G/A	-	US
GM0503	Oleur061Scf0503	499553	GTT/CTC	-	1
GM0871	Oleur061Scf0871	153993	C/G	-	11
GM0960	Oleur061Scf0960	26182	A/G	-	US
	Oleur061Scf0960	26749	A/G	-	US
	Oleur061Scf0960	27393	G/A	-	US
GM1178	Oleur061Scf1178	622380	G/T	-	5
	Oleur061Scf1178	622396	G/T	-	5
	Oleur061Scf1178	624968	TTT/CTA	-	5
	Oleur061Scf1178	628179	A/G	-	5
	Oleur061Scf1178	628964	GTATTA/AAATTC	-	5
	Oleur061Scf1178	629080	T/C	-	5
	Oleur061Scf1178	629123	TAGTG/TGGAC	-	5
	Oleur061Scf1178	629814	TGTG/CGTA	-	5
	Oleur061Scf1178	630753	GCGTGC/AAGTGT	-	5
	Oleur061Scf1178	631538	TGA/TGGA	-	5
	Oleur061Scf1178	632869	T/A	-	5
GM1459	Oleur061Scf1459	972705	C/A	-	3
GM1787	Oleur061Scf1787	58484	A/C	-	21
	Oleur061Scf1787	58605	T/C	-	21
GM2091	Oleur061Scf2091	170031	G/A	-	US
	Oleur061Scf2091	170178	C/G	-	US
	Oleur061Scf2091	170206	T/C	-	US
	Oleur061Scf2091	172566	T/G	-	US
	Oleur061Scf2091	173464	T/A	-	US
	Oleur061Scf2091	173501	T/C	-	US
	Oleur061Scf2091	173583	T/C	-	US
	Oleur061Scf2091	173729	A/G	-	US
	Oleur061Scf2091	174318	C/T	-	US
	Oleur061Scf2091	175015	TAA/TA	-	US
	Oleur061Scf2091	175071	T/A	-	US
	Oleur061Scf2091	175636	A/G	-	US
	Oleur061Scf2091	175811	C/G	-	US
	Oleur061Scf2091	175966	T/A	-	US
	Oleur061Scf2091	176408	A/C	-	US
	Oleur061Scf2091	176444	T/C	-	US
	Oleur061Scf2091	176468	G/A	-	US
	Oleur061Scf2091	176475	A/C	-	US
GM2874	Oleur061Scf2874	871356	A/G	YES	5
GM3270	Oleur061Scf3270	126286	TC/AA	-	7
GM3346	Oleur061Scf3346	640676	G/A	YES	8
GM3361	Oleur061Scf3361	582095	A/G	YES	20
	Oleur061Scf3361	582217	A/G	YES	20
	Oleur061Scf3361	582274	TTGT/TT	YES	20
	Oleur061Scf3361	582298	A/C	YES	20
	Oleur061Scf3361	582368	T/A	-	20
	Oleur061Scf3361	582444	T/A	-	20
	Oleur061Scf3361	582495	A/T	-	20
GM3663	Oleur061Scf3663	386986	G/A	YES	8
	Oleur061Scf3663	387078	T/C	-	8
GM3825	Oleur061Scf3825	171174	T/C	-	7
GM4112	Oleur061Scf4112	400436	C/T	-	13
	Oleur061Scf4112	400456	C/A	-	13
	Oleur061Scf4112	400466	G/A	-	13
	Oleur061Scf4112	422150	C/G	NO	3
	Oleur061Scf4112	425723	GA/TG	-	3
GM4351	Oleur061Scf4351	26325	G/C	-	15
GM4403	Oleur061Scf4403	193259	T/C	NO	14
GM4462	Oleur061Scf4462	60219	T/C	-	7
GM4491	Oleur061Scf4491	299627	C/G	-	18
	Oleur061Scf4491	300578	C/G	-	18
	Oleur061Scf4491	300597	G/A	-	18
GM4878	Oleur061Scf4878	118649	C/G	YES	4
	Oleur061Scf4878	118890	T/C	YES	4
GM4977	Oleur061Scf4977	92722	T/G	-	1
GM5420	Oleur061Scf5420	23710	C/T	NO	16
	Oleur061Scf5420	27925	C/T	-	16
	Oleur061Scf5420	29878	C/T	-	16
	Oleur061Scf5420	220407	A/G	-	16
GM5641	Oleur061Scf5641	83225	A/C	YES	11
	Oleur061Scf5641	84939	T/C	-	11
GM6972	Oleur061Scf6972	73917	A/G	YES	21
GM7206	Oleur061Scf7206	20070	CACG/CGCA	-	16
GM7731	Oleur061Scf7731	48216	TTT/CTC	-	8
	Oleur061Scf7731	48236	G/A	-	8
GM8230	Oleur061Scf8230	14454	T/C	-	15

^1^ GMs clustered according to their close proximity and often confirmed to be inherited as haplotypes. The GM clusters confirmed by studying the segregating progeny are shown as blue text in bold. The unconfirmed clusters are depicted by red text. The unanalyzed clusters are denoted by normal black text. * *Olea europaea var. sylvestris* genome was used as reference to determine the chromosomal location [24]. US = unfound sequence in the wild genome.

**Table 2 plants-12-00155-t002:** The mean fruit dry weight of the selected extreme phenotype progeny trees of the “Frantoio” x “Picual” cross from two seasons.

Tree Reference Number Row Tree		Fruit Dry Weight Average of Seasons 2019–2020 (g)
Frantoio		1.24
Picual		1.45
Heaviest fruit trees:		
279	59	2.04
281	4	2.03
281	29	1.84
280	39	1.77
283	12	1.77
282	35	1.70
281	15	1.68
281	13	1.67
281	17	1.65
282	32	1.64
279	42	1.63
281	55	1.60
280	47	1.57
282	46	1.56
282	11	1.55
282	33	1.51
280	7	1.50
280	13	1.49
279	19	1.41
280	45	1.39
Lightest fruit trees:		
280	8	1.13
281	16	1.11
281	42	1.08
281	34	1.07
279	34	1.04
282	48	1.01
280	27	0.99
282	30	0.97
280	43	0.97
280	26	0.93
279	16	0.91
279	35	0.90
280	59	0.88
279	37	0.87
281	39	0.81
280	4	0.81
279	32	0.80
280	30	0.68
279	33	0.55
283	11	0.43

## Data Availability

The datasets generated in the present study are available at NCBI as BioProject: PRJNA870905.

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
