# Peer review of "Identification of Genetic Markers and Genes Putatively Involved in Determining Olive Fruit Weight"

_plants, 2022, doi:10.3390/plants12010155_

Round 1

Reviewer 1 Report

Dear authors,

You haver performed a complete study to identify putative genes that could play an important role in olive fruit weight that could be used for marker assisted selection in breeding programs.

As functions of these genes are not proved, the title could be change as suggested in the PDF file.

The criteria to stablish a cut-off in the Manhatan plot is not clear.

It would also be interesting to include the chromosome division in the Manhatan plot and also dicuss the clusters and genes indentified to play a role in fruit development related to their chromosome position and with other previous works, as the one of Kaya et al (2019).

Some other suggestions and corrections are included in the PDF file.

Kind regards

Reviewer 2 Report

Review on Moret et al. Genetic markers and genes involved in determining olive fruit 2 weight

L37: A considerable amount of information is missing from the Introduction. The authors started to speak about wild species but first, it should be stated where is the place of origin of this species, where wild species grow, and list some of the most important species.

A systematic screening of scientific papers identifying QTLs in association with the fresh weight of olive as well as identifying a range of factors modifying berry weight in this species should be carried out! Such a factor is the fruit setting ratio (crop size) and it should have been described how this factor was ensured not to influence the outcome of the analysis. The Introduction part is very shallow with some repetitive elements.

The genome sequence of the olive should be mentioned.

A real-time PCR verification would be important to accept the expression profile of the identified genes.

Results

Saying the female parent has smaller fruit than the male has, is not enough for such a study. Please give the precise average fruit weight with deviations value.

L166-176: The description is rather dubious. It is not clear what it means for example „is a gene of drought tolerance” or „a gene involved in ethylene formation and anthocyanidin biosynthesis”. It should be given more accurately what kind of genes are those mentioned, what is the name if they are enzymes which reaction is catalyzed by this enzyme etc. All other elements of information that are listed here should be placed in the Discussion section and give the cited references that state the gene is involved in drought tolerance somehow.

The discussion is in general very scarce but the part dealing with the identified genes is severely deficient in terms of discussion. L350-357 should give data from other studies showing if identified genes are having something similar physiological roles in other plants. I cannot understand how such sentences (and many others here) could be written without a single reference cited:

„GM3361 has a candidate gene that codes for a cytochrome P450 family protein that is involved in several processes, such as cell differentiation, developmental growth, the response to nitrogen compound, root development and tissue development.”

L15: correct as: large fruit size is one of the main objectives…

L30: I guess family names should not be italicized

L54: what are those high values?

L64: writing words like „local” without giving information on any geographic location does not have any meaning

L175: seemed to be the most probable one

Legend to Figure 5. More data must be included: how the gene expression profile was determined?

L297: this statement is given here after many times the same statement was made.

L347: give citation here.

L374 and many other sentences: citations to relevant papers are severely missing!

L382: „This GMs” must be corrected

L409: I guess the reference genome should be cited correctly

Round 2

Reviewer 2 Report

I agree with the author in that we have different views on a number of aspects of this study. I feel the authors should further increase its meaning and reliability and making the introduction more logic but I can accept their standpoint.